# The Effect of Non-Pharmacological and Pharmacological Interventions on Measures Associated with Sarcopenia in End-Stage Kidney Disease: A Systematic Review and Meta-Analysis

**DOI:** 10.3390/nu14091817

**Published:** 2022-04-27

**Authors:** Daniel S. March, Thomas J. Wilkinson, Thomas Burnell, Roseanne E. Billany, Katherine Jackson, Luke A. Baker, Amal Thomas, Katherine A. Robinson, Emma L. Watson, Matthew P. M. Graham-Brown, Arwel W. Jones, James O. Burton

**Affiliations:** 1York Trials Unit, Department of Health Sciences, University of York, York YO10 5DD, UK; 2Department of Cardiovascular Sciences, University of Leicester, Leicester LE1 7RH, UK; r.billany@leicester.ac.uk (R.E.B.); katherine.robinson@cardiov.ox.ac.uk (K.A.R.); emma.watson@leicester.ac.uk (E.L.W.); mgb23@leicester.ac.uk (M.P.M.G.-B.); jb343@leicester.ac.uk (J.O.B.); 3NIHR Applied Research Collaboration, Leicester Diabetes Centre, University of Leicester, Leicester LE5 4PW, UK; t.j.wilkinson@leicester.ac.uk; 4Leicester Medical School, University of Leicester, Leicester LE1 7HA, UK; tb259@student.le.ac.uk (T.B.); kj120@student.le.ac.uk (K.J.); 5Department of Respiratory Sciences, College of Life Sciences, University of Leicester, Leicester LE1 7RH, UK; lab69@leicester.ac.uk; 6NIHR Leicester Biomedical Research Centre, University Hospital of Leicester NHS Trust, University of Leicester, Leicester LE5 4PW, UK; 7University Hospital of Leicester NHS Trust, Leicester LE1 5WW, UK; amalthomas@outlook.com; 8Central Clinical School, Monash University, Melbourne 3004, Australia; arwel.jones@monash.edu; 9School of Sport, Exercise and Health Sciences, Loughborough University, Loughborough LE11 3TU, UK

**Keywords:** end-stage kidney disease, dialysis, transplant, systematic review, meta-analysis, exercise, nutrition

## Abstract

This systematic review and meta-analysis provides a synthesis of the available evidence for the effects of interventions on outcome measures associated with sarcopenia in end-stage kidney disease (ESKD). Thirteen databases were searched, supplemented with internet and hand searching. Randomised controlled trials of non-pharmacological or pharmacological interventions in adults with ESKD were eligible. Trials were restricted to those which had reported measures of sarcopenia. Primary outcome measures were hand grip strength and sit-to-stand tests. Sixty-four trials were eligible (with nineteen being included in meta-analyses). Synthesised data indicated that intradialytic exercise increased hand grip strength (standardised mean difference, 0.58; 0.24 to 0.91; *p* = 0.0007; *I*^2^ = 40%), and sit-to-stand (STS) 60 score (mean difference, 3.74 repetitions; 2.35 to 5.14; *p* < 0.001; *I*^2^ = 0%). Intradialytic exercise alone, and protein supplementation alone, resulted in no statistically significant change in STS5 (−0.78 s; −1.86 to 0.30; *p* = 0.16; *I*^2^ = 0%), and STS30 (MD, 0.97 repetitions; −0.16 to 2.10; *p* = 0.09; *I*^2^ = 0%) performance, respectively. For secondary outcomes, L-carnitine and nandrolone-decanoate resulted in significant increases in muscle quantity in the dialysis population. Intradialytic exercise modifies measures of sarcopenia in the haemodialysis population; however, the majority of trials were low in quality. There is limited evidence for efficacious interventions in the peritoneal dialysis and transplant recipient populations.

## 1. Introduction

Sarcopenia, originally believed to be a condition related to age, is the term used to indicate a progressive reduction in muscle strength, quantity or quality, and function, and is now considered a muscle disease [1]. It is now recognised as being associated with a number of catabolic diseases. One of these diseases which can expedite changes in measures related to sarcopenia is chronic kidney disease (CKD). Sarcopenia is reported as a common comorbidity in individuals with CKD, with a prevalence of around 10% in non-dialysis-dependent individuals [2,3], and increasing up to 37% in those individuals with end-stage kidney disease [4]. The presence of sarcopenia in individuals with CKD is associated with low quality of life, major adverse cardiovascular events, and mortality [2,5]. The underlying mechanisms of sarcopenia in CKD are believed to revolve around the concomitant loss of strength and muscle mass [6]. The cause of this in the CKD population is multifactorial, and numerous, but negative protein balance, sedentary behaviour, physical inactivity, metabolic acidosis, inflammation, anorexia, and disturbed appetite regulation all play a role [3,7]. The loss of muscle mass and strength is more common in individuals with end-stage kidney disease (ESKD) compared to individuals with less advanced kidney disease [8,9].

There is currently a lack of effective interventions for the treatment of sarcopenia, particularly in the ESKD population. However, a previous clinical practice guideline has provided strong recommendations for exercise as the primary treatment of sarcopenia [10]. The evidence for other non-pharmacological interventions such as nutritional is less clear [11]. Currently, there are no specific drugs approved for the treatment of sarcopenia; however, recently there has been a growing interest in new therapeutic approaches in the CKD population [12]. Therefore, the aim of this systematic review (and meta-analysis) was to investigate the effect of non-pharmacological and pharmacological interventions on outcome measures associated with sarcopenia (as defined by the European Working Group on Sarcopenia in Older People (EWGSOP) [1]) in the ESKD population.

## 2. Materials and Methods

### 2.1. Protocol Registration

Methods were prespecified and documented in a protocol that was registered on International Prospective Register of Systematic Reviews; www.crd.york.ac.uk/PROSPERO (PROSPERO) with the identifier CRD42020199301.

### 2.2. Settings and Trial Population

Individuals with ESKD who have received a transplant, or are receiving dialysis (haemodialysis and peritoneal dialysis) or conservative management (for those with an estimated glomerular filtration rate <15) over the age of 18 years were included.

### 2.3. Intervention

Trials were considered eligible if they contained non-pharmacological (for the purpose of this review, these were defined as either containing diet, exercise, or lifestyle components) or pharmacological interventions (e.g., growth hormone, combined oestrogen-progesterone, dehydroepinadorsterone).

### 2.4. Comparison

Any concurrent control group who is receiving usual care could serve as the control. Control groups that receive usual care or a placebo (for dietary or pharmacological interventions), or who did not receive an intervention designed to modulate sarcopenia were included. Exercise trials that had included active control groups (e.g., stretching) were excluded, as were trials of acute interventions.

### 2.5. Outcome

Recently, the European Working Group on Sarcopenia in Older People (EWGSOP) published a consensus paper [1] highlighting a number of outcome measures to assess, confirm, and determine severity of sarcopenia. The outcomes in this review were chosen as a result of their inclusion in this paper. The primary outcome was muscle strength (hand grip strength (HGS) and the following sit-to-stand tests (STS), 5, 30, and 60). The secondary outcomes were muscle quality and quantity (assessed by magnetic resonance imaging (MRI), dual-energy X-ray absorptiometry (DEXA), bioelectrical impedance analysis (BIA), and computed tomography (CT) imaging), physical performance (assessed by the short physical performance battery (SPPB), the timed-up-and-go test (TUG), 400 m walk test, and gait speed), and sarcopenia health-related quality of life as assessed by the SARQoL questionnaire.

### 2.6. Trial Design

Trials included in this review had to have adhered to the following trial designs: parallel-group randomised controlled trials (allocation at individual or cluster levels) or crossover randomised trials.

### 2.7. Search Strategy

Searches were conducted to identify any relevant completed or ongoing systematic reviews using the following resources: Cochrane, PROSPERO, and the National Health Service Centre for Reviews and Dissemination (Health Technology Assessment (HTA) and Database of Abstracts of Reviews of Effects (DARE)). The following bibliographical databases and trial registers were searched for completed and ongoing trials: MEDLINE, EMBASE, CINAHL, Cochrane Central Register of Controlled Trials (CENTRAL), ClinicalTrials.gov, and the ISCRTN Registry. British Library (ETHOS), OpenGrey, and Conference Proceedings Citation Index (Web of Science™ Core Collection) were searched for unpublished data. All databases were searched from inception to 19 July 2021, and no limits on language were set. Database searches were supplemented with internet searches (e.g., Google Scholar), and contact with the Physical Activity and Wellbeing Kidney Research Study Group (in the United Kingdom). An example of a full search strategy for MEDLINE, EMBASE, and CINAHL databases is presented in Appendix A. Other databases were searched by using different combinations Wof these search terms. Search results were compiled using the web-based screening and data extraction tool Covidence (Veritas Health Innovation Ltd., Melbourne, Australia) as recommended by the Cochrane Collaboration. Duplicate citations were removed, and title and abstracts were screened independently by two reviewers against the inclusion criteria (if there was disagreement, Wthen this was settled through the use of a third reviewer). Full-text articles of trials not excluded based on title or abstracts were retrieved and assessed by two reviewers. Conference abstracts and trials included on registries only (e.g., ClinicalTrials.gov) were excluded.

### 2.8. Selection Criteria, Data Extraction, and Quality Appraisal

We developed, tested, and refined a structured data collection form based on the Cochrane Data Extraction Template for interventions. For each included trial, information on trial methods, participants, interventions/comparator, and outcomes was extracted and cross-checked by one reviewer (DSM). Risk of bias for each trial was assessed using the Cochrane Risk of Bias Tool across five domains. Each domain was classified as adequate, unclear, or inadequate, with risk of bias for each trial to be classified using the following criteria: (1) low risk of bias (all criteria are deemed adequate), (2) moderate risk of bias (one criterion graded as inadequate or two graded as unclear), and (3) high risk of bias (more than one criterion is deemed inadequate, or more than two are graded unclear). Funnel plots were used to visually assess publication bias in the meta-analyses performed for the primary outcome only. Formal testing for plot asymmetry would only be performed where the meta-analysis contains more than ten trials [13].

### 2.9. Data Synthesis

Where means and standard deviation of outcome measures were not available, they were estimated from medians and interquartile ranges [14]. Gait speed data were converted from cm/s to m/s for one trial [15], and were provided by the authors for another [16]. HGS was converted from lbs to kg for one trial [17]. Data for mid-arm muscle area (MAMA) were subtracted for one trial [18] using Web-Plot Digitizer version 4.5 [19] and 95% confidence intervals were converted to standard deviations [13]. A meta-analysis was performed for trials that reported the same outcome measures using a generic inverse variance random effects method via Review Manager (RevMan) version 5.3.26 (The Cochrane Collaboration, 2020). Primary and secondary measures of efficacy were treated as continuous data and interpreted as either difference in means or standardised mean difference dependent on the methods of measurement. Analysis was based on the final (post-intervention) values only (at last follow-up) with the exception of mean change data from two trials [15,20]. Statistical heterogeneity was interpreted using the *I*^2^ value. Data were not pooled (or subgroup analysis was considered) if *I*^2^ > 40% (this is the threshold to which heterogeneity is considered important). Separate analysis was performed for each type of population (dialysis and transplant) and each non-pharmacological and pharmacological intervention. We had prospectively planned a network meta-analysis (NMA); however, this was not possible as a result of a limited number of trials for each population reporting the same sarcopenia-associated outcome. In addition, variances between the delivered interventions within the included trials suggested that the transitivity assumption (needed for NMA) was unlikely to be met.

## 3. Results

### 3.1. Characteristics of Included Trials

Figure 1 provides a flow diagram of trial selection. Sixty-four trials were eligible for the review (Table 1, Table 2 and Table 3), with 19 trials being included in meta-analyses. Eleven conference abstracts were excluded at the full-text screening stage (due to insufficient information). There were 54 trials in the dialysis population (43 in the haemodialysis, 7 in the peritoneal dialysis, and 4 trials containing both dialysis populations) (Table 1 and Table 2). In total, 23, 20, and 8 trials tested exercise, nutritional supplement, and pharmacological interventions, respectively. Two trials tested both exercise and pharmacological interventions [15,21], and one trial tested an exercise and a nutritional intervention [22]. There were ten eligible trials in the transplant recipient population (Table 3). The most prevalent measurements of muscle strength, muscle quality/quantity, and physical performance in the ESKD population were HGS (*n* = 26), lean whole body mass (LBM) (*n* = 29), and gait speed (*n* = 15), respectively. There were no trials identified that included conservative management participants, and no trial reported the SARQoL questionnaire as an outcome (Table 1, Table 2 and Table 3). Twenty-nine trials (45%) reported an *a* priori power calculation.

### 3.2. Risk of Bias

Risk of bias summaries are provided in Figure A1, Figure A2 and Figure A3. Only 10 (16%) of the included trials were rated as having an overall low risk of bias. Funnel plots are provided in Figure A4 (for the analyses presented in Figure 2, Figure 3, Figure 4 and Figure 5). There was no observation of publication bias.

### 3.3. Muscle Strength

#### 3.3.1. Hand Grip Strength

Exercise Interventions

Eight trials reported measurement of HGS [17,25,29,30,34,38,39,41] following programmes of intradialytic exercise, with data available from seven trials (all except [41]). The synthesised data showed (254 participants) a statistically significant increase in HGS (standardised mean difference (SMD), 0.58; 0.24 to 0.91; *p* = 0.0007; *I*^2^ = 40%) (Figure 2). Four trials [26,29,35,37] reported data on HGS following exercise programmes taking place outside of dialysis, although there was considerable heterogeneity (*I*^2^ = 89%). Two of these trial reported statistically significant increases [26,35], and two reported no significant changes [29,37] (Table 4). One trial in the peritoneal dialysis population [40] reported no changes in HGS following an exercise intervention. There was significant heterogeneity (*I*^2^ = 75) between trials (28 participants) investigating the effect of programmes of exercise on HGS [71,73] in transplant recipients, with one trial reporting a significant increase [71]. A further trial showed no effect of a lifestyle intervention [70].

2.Nutritional Interventions

Data from two trials (110 participants) [53,64] investigating the effect of Vitamin D (cholecalciferol) on HGS were available, but there was considerable heterogeneity between the trials (*I*^2^ = 60%). Neither trial [53,64] reported any significant change with Vitamin D. Other interventions including L-carnitine [18,67] and keto acid supplementation [58] appeared to have no effect in the dialysis population.

3.Pharmacological Interventions

Three trials reported measuring HGS following the administration of growth hormone [48,55,56] in the haemodialysis population, but the data were not suitable for meta-analysis. Individual data from two of these trials showed no statistically significant increase [48,55]. Two trials investigated the effect of anabolic steroid supplementation on HGS, one reported a significant increase [63], whilst there was no change reported in the other [20].

#### 3.3.2. Sit-to-Stand

Exercise Interventions

Synthesised data from six trials (212 participants) [14,15,16,17,21,30] indicated that intradialytic exercise resulted in no statistically significant change in STS5 score (mean difference (MD), −0.78 s; 95% confidence interval, −1.86 to 0.30; *p* = 0.16; *I*^2^ = 0%) (Figure 3). For STS60 score, intradialytic exercise (data from seven trials (425 participants) [16,17,21,27,30,38,42]) resulted in a statistically significant increase (MD, 3.74 repetitions; 2.35 to 5.14; *p* < 0.0001; *I*^2^ = 0%) (Figure 4). A further trial in 296 dialysis participants showed that a programme of home-based walking significantly increased STS5 score compared to a control group [31]. For the peritoneal dialysis population, one trial [23] reported no statistically significant change in STS30 following a programme of exercise. Data from two trials (62 participants) [69,71] was available in the transplant population investigating the effect of programmes of exercise on STS60; however, there was considerable statistical heterogeneity between trials (*I*^2^ = 83%). Individually both trials reported statistically significant increases in STS60 (only for the resistance group in one trial [69]).

2.Nutritional Interventions

Synthesised data from two trials [22,66] in the haemodialysis population (98 participants) indicated that oral whey protein supplementation resulted in no statistically significant change in STS30 score (MD, 0.97 repetitions; −0.16 to 2.10; *p* = 0.09; *I*^2^ = 0%) (Figure 5). There was no significant effect of pomegranate extract [68] or beta-hydroxy-beta-methylbutyrate [49] supplementation on STS30.

3.Pharmacological Intervention

One trial reported a lack of effect of anabolic steroids on STS5 [15].

### 3.4. Muscle Quality/Quantity

#### 3.4.1. Exercise Interventions

Data from four trials [14,15,21,32] reported measurement of LBM using DEXA following intradialytic exercise. Synthesised data from three trials [14,15,21] (70 participants) reported a non-statistically significant effect (MD, 0.63 kg; −3.46 to 4.72; *p* < 0.76; *I*^2^ = 0%) (Figure 6). Mean change data for mid-thigh cross-sectional area (MT-CSA) ([21,24] and fat-free mass (FFM)) [25,28] were available from two trials each (which included programmes of intradialytic exercise); respectively, there was considerable heterogeneity between trials (*I*^2^ = 57% for MT-CSA, and *I*^2^ = 54% for FFM); neither outcome was meta-analysed. No significant changes for either of these outcomes were reported in these trials. Four trials involving programmes of exercise in the transplant recipient reported measurement of LBM [72,73,74,76], with data available from two trials [73,74] (107 participants) for synthesis, although there was considerable heterogeneity (*I*^2^ = 78%); resultantly, a meta-analysis was not performed. One trial reported a statistically significant increase [73], whilst another reported no difference between the intervention and control groups [74]. Trials involving lifestyle interventions of nutrition counselling and exercise/physical activity programmes [70,77] reported lack of effects on MAMA [70], LBM, [70,77], or FFM [70].

#### 3.4.2. Nutritional Interventions

Synthesised data from two trials [18,43] including 108 haemodialysis participants indicated that L-carnitine supplementation significantly increased MAMA (MD, 3.10 cm^2^; 0.92 to 5.28; *p* = 0.005; *I*^2^ = 0%) (Figure 7). One of these trials [18] also reported data for LBM, skeletal muscle mass, and appendicular lean mass (ALM) with no statistically significant change in these outcomes following L-carnitine supplementation. Synthesised data from two trials [22,66] in the haemodialysis population (98 participants) indicated that oral whey protein supplementation resulted in no statistically significant effect on LBM (MD, −1.55 kg; −4.25 to 1.14; *p* = 0.26; *I*^2^ = 0%) (Appendix A). Data were reported on LBM from trials investigating a number of heterogeneous nutritional interventions (see Table 2). Individual results from these trials reported statistically significant increases in LBM following water-soluble vitamin supplementation [44], amino acid supplementation [54], and creatine supplementation [60]. Other trials reported data for ALM [49], MAMA [44], and FFM [45,51,62] and individually reported no significant changes (see Table 2 for interventions). For the peritoneal dialysis population, data from three trials were available reporting on the effect of protein supplementation on mid-arm muscle circumference (MAMC) [50,61,65]; there was heterogeneity between trials (*p* = 45%). One trial reported a statistically significant increase in MAMC (along with LBM) [65], whilst there was no change for this variable in the other two trials [50,61].

#### 3.4.3. Pharmacological Interventions

Synthesised mean change data from two trials [15,20] investigating the effect of nandrolone decanoate (an anabolic steroid) on LBM showed a statistically significant increase (MD, 3.10 kg; 2.12 to 4.08; *p* < 0.044; *I*^2^ = 0%) (Figure 8). One of these trials [15] also reported a significant increase in MT-CSA, and another has shown an increase in FFM following oxymetholone [63]. Mean change data were available for LBM from three trials [48,52,57] investigating the effect of growth hormone. There was considerable heterogeneity between trials (*I*^2^ = 75%). Two trials reported significant increases in LBM following growth hormone injections compared to placebo [48,52]. In two trials investigating the effect of early steroid withdrawal in transplant recipients there was no effect of this on LBM [75,78].

### 3.5. Physical Performance

#### 3.5.1. Gait Speed

Exercise Interventions

Eight trials [14,15,16,21,25,27,33,41] reported measurement of gait speed [16]. Data were available for synthesis from five trials (364 participants) [15,16,25,27,33]; there was a significant increase in gait speed following intradialytic exercise (SMD, 0.24; 0.03 to 0.44; *p* = 0.03; *I*^2^ = 0%) (Figure 9). In transplant recipients, one trial [70] found no effect of a lifestyle intervention on gait speed.

2.Nutritional Interventions

Synthesised data from two trials [22,66] (98 participants) indicated that oral whey protein resulted in no significant effect on gait speed (MD, 0.08 m/s; −0.02 to 0.18; *p* = 0.12; *I*^2^ = 0%) (Figure 10). Two trials showed no effect of creatine supplementation [60] or beta-hydroxy-beta-methylbutyrate supplementation [49] on gait speed.

3.Pharmacological Interventions

Data (which were unsuitable for meta-analysis) were reported for two trials investigating the effect of human growth hormone [48,55]. Only one trial reported a significant increase in gait speed following the administration of growth hormone [55]. Another trial found a lack of effect following anabolic steroid supplementation [15].

#### 3.5.2. Timed-Up-and-Go and Short Physical Performance Battery

Exercise Intervention

Synthesised data from two trials (69 haemodialysis participants) [29,33] for TUG reported no significant effect (MD, −1.05 s; −2.12 to 0.02; *p* = 0.06; *I*^2^ = 0%) (Figure 11) following intradialytic exercise. Moreover, a supervised programme of exercise performed on non-dialysis days significantly improved TUG [26]. Programmes of home-based walking [29,36] and intradialytic exercise [16] did not significantly improve SPPB [16,36] or TUG [29]. In contrast, one trial [23] in the peritoneal dialysis population and another in transplant recipients [71] demonstrated significant increases in TUG following programmes of exercise.

2.Nutritional Intervention

Synthesised data from two trials [22,66] (98 participants) indicated that oral whey protein resulted in no change in TUG (MD, −0.54 s; −1.33 to 0.25; *p* = 0.18; *I*^2^ = 0%) (Appendix A).

## 4. Discussion

This is the first review that has aimed to synthesise the effect of non-pharmacological and pharmacological interventions for sarcopenia outcomes (using the most up-to-date and widely accepted definition [1]) in the ESKD population. The main findings of this review were that intradialytic exercise significantly improved measures of muscle strength (HGS and STS60) and physical performance as measured by gait speed. However, the majority of trials included in the review were considered to be at high risk of bias. There was some evidence that programmes of exercise in transplant recipients may improve STS scores. The evidence for nutritional and pharmacological interventions was less clear, with some tentative evidence that L-carnitine and nandrolone decanoate may have favourable effects on muscle quantity (MAMA and LBM, respectively) in individuals receiving haemodialysis. There was a lack of evidence for efficacious interventions to treat sarcopenia in the transplant and peritoneal dialysis population, and there were no included trials in those individuals with ESKD receiving conservative management.

A recent systematic review exploring the effect of exercise interventions on objective physical function in the ESKD population [79] reported that the majority of included trials reported a significant improvement in STS and HGS, although unlike the present review they were not able to perform a meta-analysis for these outcomes. This is in agreement with another review [80] that demonstrated that exercise training in the haemodialysis population was able to increase muscle strength. Our review confirms that exercise is efficacious at modifying outcomes associated with sarcopenia; however, the evidence for pharmacological and nutritional interventions is less clear. This review included trials with a number of heterogeneous nutritional and pharmacological interventions with a lack of evidence for their efficacy on measures of sarcopenia. However, this is with the exception of synthesised data for L-carnitine and nandrolone-decanoate showing modifications to MAMA and LBM. However, it is unclear whether changes to these outcomes would translate to improvement in muscle strength and function.

Sarcopenia is highly prevalent in CKD [3], particularly for those with the advanced stages of the disease (ESKD) [6]. It is associated with hard endpoints including cardiovascular events and mortality [2,5]. With prevalence of ESKD projected to increase [81], identifying effective interventions for the treatment of sarcopenia is particularly relevant. Therefore, the finding of this review, that intradialytic exercise improves HGS and gait speed, has clinical significance. A low walk (gait) speed has been shown to be associated with mortality in 752 individuals receiving dialysis [82], with a walk speed of >0.6 m/s associated with greater survival [82]. Another study [83] has also reported that both low gait speed and HGS are predictors of cardiovascular events and all-cause mortality in individuals receiving haemodialysis [83]. This supports the recent shift from low muscle mass to low muscle strength as a key characteristic for the diagnosis of sarcopenia [1], as low muscle strength appears to be better at predicting outcomes [3,84]. Furthermore, muscle strength (STS and HGS) can be easily evaluated in the clinical setting (outpatient clinics and dialysis units, etc.). The evidence from this review that intradialytic exercise increases muscle strength, coupled with recent RCT data [16] (that this mode of exercise improves cardiovascular health and is safe), suggests that the methods of implementation should be considered as outlined in the recent Clinical Practice Guideline for Exercise and Lifestyle in CKD [85].

It is believed that increasing protein intake may be an effective countermeasure to sarcopenia for individuals with CKD. This is highlighted by the recommendation of increased intake (compared to the general population) for individuals with ESKD in the updated KDOQI Clinical Practice Guideline for Nutrition in CKD [86]. However, the present review found limited current RCT evidence for the efficacy of protein supplementation for sarcopenia in CKD, a point that has recently been highlighted by others [6]. Protein without an adequate exercise stimulus often provides little benefit, although notably the largest RCT to date in the ESKD population investigating the combined effect of exercise and protein supplementation found no effect on muscle strength or function [22]. This review identified a limited number of trials in the peritoneal dialysis and transplant recipient population. Given the positive effects that we have seen for exercise interventions (particularly for muscle strength in the haemodialysis population), it would be prudent to test these in future RCTs involving other ESKD populations. A recent review article [6] has highlighted a number of pharmacological interventions as having the potential to mitigate sarcopenia in the CKD population. However, this review found no evidence for the benefit of pharmacological interventions on muscle strength. There was some indication from synthesised data that nandrolone-decanoate increases LBM and individual data from two trials show that growth hormone may improve LBM. Whether these changes may improve outcome is unlikely. A previous trial of nandrolone decanoate in individuals with rheumatoid arthritis found an increase in LBM but no accompanying change in muscle strength [87]. Properly powered (<50% of the included trials reported an *a priori* sample size calculation) trials are required to test both the efficacy and safety of pharmacological and nutritional interventions in the ESKD population. This should enable a wide range of evidence-based therapeutics to be available in line with a personalised medicine approach to tackling sarcopenia. Lastly, although we have shown that exercise programmes may be an effective countermeasure to sarcopenia in the ESKD population, there remains a lack of evidence for these interventions on associated hard endpoints such as cardiovascular events and mortality. Despite the inclusion of 64 trials in the review, only a small number of these were able to be included in meta-analyses (with only fifteen trials being included in analyses for the primary outcome (muscle strength)) and the majority were assessed as having a high risk of bias.

## 5. Conclusions

Currently, exercise appears to be the strongest therapeutic intervention for sarcopenia in the end-stage kidney disease population. There is a lack of proven efficacy for nutritional and pharmacological interventions.

## Figures and Tables

**Figure 1 nutrients-14-01817-f001:**
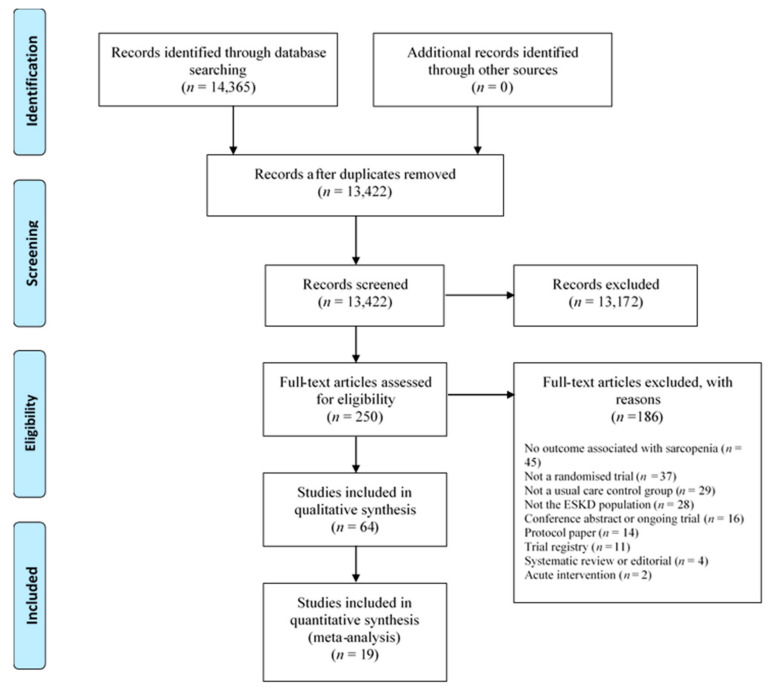
Prisma flow diagram of trial selection. ESKD = end stage kidney disease.

**Figure 2 nutrients-14-01817-f002:**
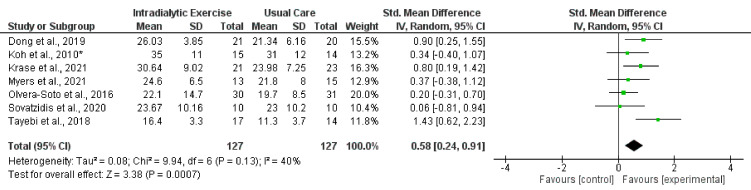
Effect of intradialytic exercise on grip strength in individuals receiving haemodialysis. Data are expressed as standardised mean difference and 95% CI. * Data for exercise and control groups only [17,25,29,30,34,38,39].

**Figure 3 nutrients-14-01817-f003:**
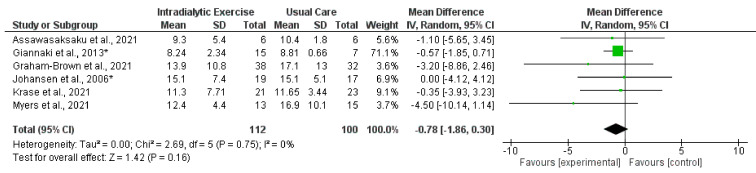
Effect of intradialytic exercise on sit-to-stand test 5 (seconds) in individuals receiving haemodialysis. Data are expressed as mean difference and 95% confidence interval (CI). * Data for exercise and control groups only [14,15,16,17,21,30].

**Figure 4 nutrients-14-01817-f004:**
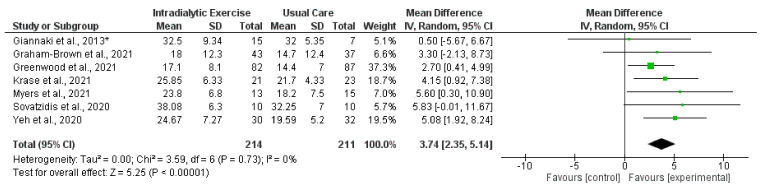
Effect of intradialytic exercise on sit-to-stand test 60 (repetitions) in individuals receiving haemodialysis. Data are expressed as mean difference and 95% CI. * Data for exercise and control groups only [16,17,21,27,30,38,42].

**Figure 5 nutrients-14-01817-f005:**
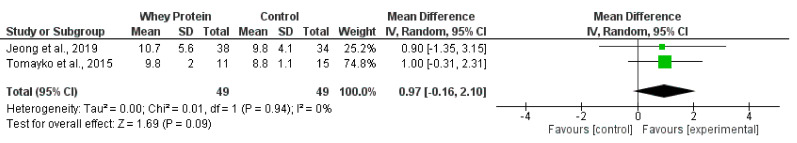
Effect of whey protein supplementation on sit-to-stand test 30 (repetitions) in individuals receiving haemodialysis. Data are expressed as mean difference and 95% CI. Data for whey protein and control groups from both trials [22,66].

**Figure 6 nutrients-14-01817-f006:**
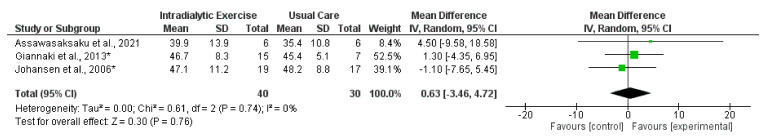
Effect of intradialytic exercise on lean whole body mass (kg) measured by DEXA in individuals receiving haemodialysis. Data are expressed as mean difference and 95% CI. * Data for exercise and control groups only [14,15,21].

**Figure 7 nutrients-14-01817-f007:**
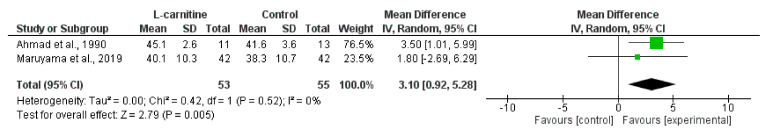
Effect of L-carnitine supplementation on mid-arm muscle area in individuals receiving haemodialysis. Data are expressed as mean difference and 95% CI [18,43].

**Figure 8 nutrients-14-01817-f008:**
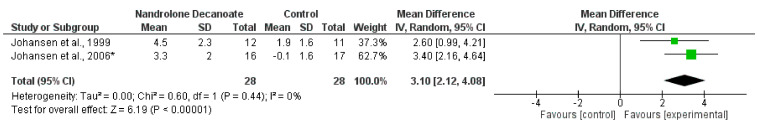
Effect of nandrolone decanoate on lean whole body mass (mean change data) measured by DEXA in haemodialysis patients. Data are expressed as mean difference and 95% CI. * Data for nandrolone decanoate and control groups only [15,20].

**Figure 9 nutrients-14-01817-f009:**
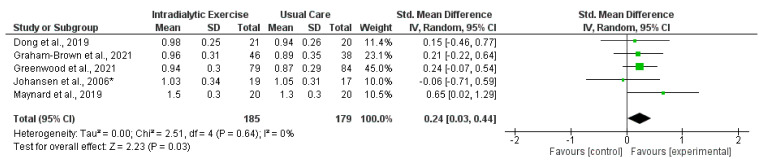
Effect of intradialytic exercise on gait speed (m/s) in individuals receiving haemodialysis. Data are expressed as standardised mean difference and 95% CI. * Data for exercise and control groups only [15,16,25,27,33].

**Figure 10 nutrients-14-01817-f010:**
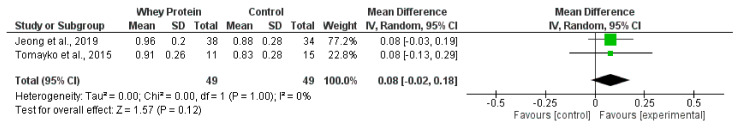
Effect of whey protein supplementation on gait speed (m/s) in individuals receiving haemodialysis. Data are expressed as mean difference and 95% CI. Data for whey protein and control groups from both trials [22,66].

**Figure 11 nutrients-14-01817-f011:**
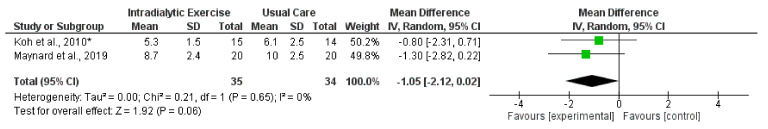
Effect of intradialytic exercise on timed-up-and-go score (s) in individuals receiving haemodialysis. Data are expressed as mean difference and 95% CI. * Data for exercise and control groups only [29,33].

**Table 1 nutrients-14-01817-t001:** Characteristics of exercise trials in the peritoneal dialysis and haemodialysis population that reported an outcome associated with sarcopenia.

Trial	Country	Trial Design	Age; Sex	Haemodialysis or Peritoneal Dialysis	Sample Size (*n* = Randomised)	Dialysis Vintage	Type of Intervention	Intervention Description (Method of Delivery, Dose, Frequency, Duration)	Intervention Compliance	Type of Comparison	Length of Follow-Up	Sarcopenia- Related Outcomes	Prospective Power Calculation Reported
Assawasaksakul et al., 2021 [14]	Thailand	Parallel group RCT	Intervention = 52.5 ± 12.9 years; 33.3% male. Control = 53.7 ± 17.2 years; 50% male.	Haemodialysis	12	Intervention = 105 (30.0,155.3) months. Control = 66.5 (20.0, 89.8). Date presented as median (IQR).	Intradialytic aerobic exercise programme.	3 × week. Participants performed cycling exercise training for 60 min during the first 2 h of each dialysis session using a cycle ergometer at an RPE up to 12.	Not reported	Usual care.	6 months	Gait Speed, LBM, STS5	Not reported
Bennett et al., 2020 [23]	USA	Parallel group RCT	Intervention = 57.7 ± 16.3 years; 61% male. Control = 58.3 ± 16.7 years; 46% male.	Peritoneal dialysis	36	Intervention = 18 (8, 28) months. Control = 23 (6, 48). Date presented as median (IQR).	Home-based exercise programme.	3 × week. Walking or cycling exercise (for 10–30 min). Frequency increased by 1 day per week until 300 min reached. 3–5 × week upper and lower body resistance exercises.	77%	Usual care.	3 months	STS30, TUG	Not reported
Cheema et al., 2007 [24]	Australia	Parallel group RCT	Intervention = 60.0 ± 15.3 years; 71% male. Control = 65.0 ± 12.9 years; 68% male.	Haemodialysis	49	Intervention = 3.3 (0.3, 16.7) years. Control = 1.6 [0.6,10.3]. Date presented as median (IQR).	Intradialytic resistance exercise programme.	3 × week. Upper and lower body exercises at an RPE of 15 to 17. Duration not reported.	≈80%	Usual care.	3 months	MT-CSA	Yes
Dong et al., 2019 [25]	China	Parallel group RCT	Intervention = 59 (32.5, 66.5) years; 42.9% males. Control = 62.5 (50.5, 70) years; 60%. Date presented as median (IQR).	Haemodialysis	45	Intervention = 69 (31.5, 87.5) months. Control = 57.5 (32.5, 86.5). Date presented as median (IQR).	Intradialytic resistance exercise programme.	3 × week. Upper and lower body exercises lasting 1–2 h during dialysis.	Not reported	Usual care.	3 months	Gait Speed, Hand Grip Strength, SMM, FFM	Not reported
Frih et al., 2017 [26]	Tunisia	Parallel group RCT	Intervention = 64.2 ± 3.4 years. Control = 65.2 ± 3.1 years. Sex not reported.	Haemodialysis	50	Intervention = 72.7 ± 12.7 months. Control = 73.6 ± 13.4 months.	Aerobic and resistance exercise programme on non-haemodialysis days.	4 × week. Upper and lower body exercises lasting 60 min during dialysis. Aerobic exercise included cycling and walking for 20 min at 5–6 RPE.	Not reported	Usual care.	4 months	Hand Grip Strength, STS60, TUG	Not reported
Giannaki et al., 2013 [21]	Greece	Parallel group RCT	Intervention 1 = 56.4 ± 12.5 years; 73% male. Intervention 2 = 55.7 ± 10.4 years; 57% male. Control = 56.8 ± 16.5 years; 71% male.	Haemodialysis	45	Intervention 1 = 3.9 ± 1.3 years. Intervention 2 = 4.0 ± 1.7 years. Control = 3.6 ± 1.5 years.	Intradialytic aerobic exercise programme and intradialytic aerobic exercise programme and dopamine.	Intervention 1 = 3 × week. Cycling at an intensity of 60–65% of maximal exercise capacity. Intervention 2 = 0.25 mg/dose of ropinirole (a dopamine agonist) in an empty capsule. Duration not reported.	Not reported	The control group took a plain flour placebo capsule.	6 months	Gait Speed, LBM, MT-CSA, STS5, STS30, STS60	Yes
Graham-Brown et al., 2021 [16]	UK	Parallel group RCT	Intervention 1 = 55.5 ± 15.5 years; 65% male. Control = 58.9 ± 14.9 years; 82% male.	Haemodialysis	130	Intervention = 1.2 (0.5, 3.7) years. Control = 1.3 (0.4, 3.2) years. Date presented as median (IQR)	Intradialytic aerobic exercise programme.	3 × week, for 30 min for 6 months. Cycling at an intensity of RPE 12–14.	71.7%	Usual care.	6 months.	Gait Speed, SPPB, STS5, STS60	Yes
Greenwood et al., 2021 [27]	UK	Parallel group RCT	Intervention 1 = 60.5 ± 15 years; 58% male. Control = 59.8 ± 14.1 years; 62% male.	Haemodialysis	335	Not reported.	Intradialytic aerobic and resistance exercise programme.	3 × week, for 30–40 min. 2 × week, lower extremity muscular conditioning exercises.	48.7%	Usual care.	6 months.	STS60, TUG	Yes
Groussard et al., 2015 [28]	France	Parallel group RCT	Intervention 1 = 66.5 ± 4.6. years; 63% male. Control = 68.4 ± 3.7 years; 70% male.	Haemodialysis	20	Intervention = 36.6 ± 8.2 months. Control = 41.2 ± 8 months.	Intradialytic aerobic exercise programme.	3 × week, for 30 min. Cycling at an intensity of 55–60% peak power.	Not reported	Usual care.	3 months	FFM	Not reported
Johansen et al., 2006 [15]	USA	Parallel group RCT	Intervention 1 = 55.7 ± 13.4 years; 53% male; intervention 2 = 54.4 ± 13.6 years; 60% male; intervention 3 = 55.5 ± 12.5 years; 65% male; control = 56.8 ± 13.8 years; 70% male.	Haemodialysis	79	Intervention 1 = 40 (3, 288) months. Intervention 2 = 33 (3.5, 108) months. Intervention 3 = 14 (4, 152) months. Control = 25.5 (3, 156) months. Data presented as median (IQR).	Intradialytic resistance exercise programme and nandrolone decanoate.	Intervention 1 = × 1 a week intramuscular injections of nandrolone decanoate. Intervention 2 = × 3 a week lower body resistance training during haemodialysis. Duration not reported. Intervention 3 = nandrolone injections + resistance exercise during haemodialysis.	Six participants discontinued study drug (four who were receiving placebo and two who were receiving nandrolone) before the end of the treatment period.	Control group received a placebo injection that was identical in appearance to the active drug.	3 months	Gait Speed, LBM, MT-CSA, STS5	Yes
Koh et al., 2010 [29]	Australia	Parallel group RCT	Intervention 1 = 52.3 ± 10.9 years; 66.6% male; intervention 2 = 52.1 ± 13.6 years; 73.3% male; control = 51.3 ± 14.4 years; 50% male.	Haemodialysis	70	Intervention 1 = 32.1 ± 26.7 months. Intervention 2 = 37.0 ± 31.1 months. Control = 25.8 ± 22.2 months.	Intradialytic aerobic exercise and home -based walking programme.	Intervention 1 = 3 × week, for 15–45 min. Cycling at an intensity of RPE 12–13. Intervention 2 = 3 × week unsupervised walking at RPE 12–13 for 15–45 min.	Intradialytic training = 75% ± 19%. Home- based walking = 71% ± 13%.	Usual care.	6 months	Hand Grip Strength, TUG	Yes
Krase et al., 2021 [30]	Greece	Parallel group RCT	Intervention 1 = 66.04 ± 15.35 years; 76% male. Control = 68.26 ± 11.07 years; 43% male.	Haemodialysis	48	Intervention = 7.29 ± 4.0 years. Control = 5.39 ± 5.55 years.	Intradialytic aerobic exercise programme.	3 × week, for 60 min. Ergometer cycling at an intensity of 60% peak power.	Not reported.	Usual care.	7 months	Hand Grip Strength, STS5, STS60	Yes
Manfredini et al., 2017 [31]	Italy	Parallel group RCT	Intervention 1 = 63 ± 13 years; 64% male. Control = 64 ± 12 years; 68% male.	Haemodialysis & Peritoneal dialysis	296	Not reported.	Home-based walking programme.	3 × week for 10 min	Out of 104 patients in the exercise arm who were re-evaluated after 6 months, level of adherence to the exercise program was high for 55 patients and low for 49 patients.	Usual care.	6 months	STS5	Yes
Marinho et al., 2016 [32]	France	Parallel group RCT	Intervention = 71.5 (58.5, 87.2) years; 50% males. Control = 76 (59, 83) years; 43%. Date presented as median (IQR).	Haemodialysis	14	Not reported.	Intradialytic resistance exercise programme.	3 × week of lower body resistance training at 60% 1 RM. Duration not reported.	Not reported.	Usual care	2 months	LBM	Not reported
Maynard et al., 2019 [33]	Brazil	Parallel group RCT	Intervention = 49 ± 15.2 years; 60% male. Control = 43.9 ± 11.7 years; 50% male.	Haemodialysis	45	Intervention = 62.7 ± 34.20 months. Control = 55.95 ± 38.87 months.	Intradialytic aerobic and resistance exercise programme performed with video games.	3 × week, for 30–60 min. Lower and upper body resistance exercises and ergometer cycling. At an intensity of 12–14 RPE.	Not reported.	Usual care	3 months	Gait Speed, TUG	Yes
Myers et al., 2021 [17]	USA	Parallel group RCT	Intervention = 66.3 ± 7.6years; 85% male. Control = 66.2 ± 6.7 years; 66% male.	Haemodialysis	28	Intervention = 4.25 ± 3.9 years. Control = 4.05 ± 3.9 years.	Home-based exercise programme.	7 × week, for 45 min. Aerobic and resistance exercise performed at an intensity of 12–14 RPE.	Not reported.	Usual care	3 months	Hand Grip Strength, STS5, STS60	Yes
Olvera-Soto et al., 2016 [34]	USA	Parallel group RCT	Intervention = 28.5 (23, 46) years; 47% males. Control = 29 (19, 38) years; 61%. Date presented as median (IQR).	Haemodialysis	61	Intervention = 12 (5.75, 37.7) months. Control = 18 (8, 39). Date presented as median (IQR).	Intradialytic resistance exercise programme.	2 × week, for 50 min. Upper and lower body resistance exercises.	Not reported.	Usual care	3 months	Hand Grip Strength, MAMA, MAMC	Not reported
Rosa et al., 2021 [35]	Brazil	Parallel group RCT	Intervention 1 = 53 ± 13 years; 55% male. Intervention 2 = 54 ± 10 years; 58% male. Control = 52 ± 17 years; 57% male.	Haemodialysis	266	Intervention 1 = 54.4 ± 13.8 months. Intervention 2 = 52.1 ± 11.1 months. Control = 51.7 ± 12.5 months.	Pre-dialysis dynamic and isometric resistance exercise programme.	Intervention 1 = 3 × week for 40 min. Upper and lower body exercises increasing to an RPE of 7–8. 2 = same programme as intervention 1, however they performed isometric contractions.	Not reported.	Usual care	6 months	Hand Grip Strength, FFM	Not reporter
Sheshadri et al., 2020 [36]	USA	Parallel group RCT	Intervention = 60 (53,66) years; 93% males. Control = 56 (51, 65) years; 63%. Date presented as median (IQR)	Haemodialysis & Peritoneal dialysis	60	Intervention = 3.7 (1.5, 7.2) months. Control = 1.9 (0.95, 4.7). Date presented as median (IQR).	Home-based walking programme.	Participants were provided with pedometers and were provided with weekly step goals and counselling sessions.	95%	Usual care	6 months	SPPB	Yes
Song et al., 2012 [37]	South Korea	Parallel group RCT	Intervention = 52.1 ± 12.4 years; 60% male. Control = 54.6 ± 10.1 years; 60% male.	Haemodialysis	44	Intervention = 38.9 ± 26.1 months. Control = 45.9 ± 56.2 months.	Pre-dialysis resistance exercise programme.	3 × week lasting 30 min. Consisting of upper and lower body exercises.	Not reported.	Usual care	3 months	Hand Grip Strength, MAMC, SMM	Yes
Sovatzidis et al., 2020 [38]	Greece	Parallel group RCT	Intervention = 52.8 ± 17.1 years; 80% male. Control = 53 ± 7.6 years; 90% male.	Haemodialysis	24	Not reported.	Intradialytic aerobic exercise programme.	3 × week, for 6 months. Duration was self-selected. Ergometer cycling at an intensity of RPE 11–13.	81%	Usual care	6 months	Hand Grip Strength, STS60	Yes
Tayebi et al., 2018 [39]	Iran	Parallel group RCT	Intervention = 64.4 ± 8.4 years; 71% male. Control = 63.2 ± 11.6 years; 50% male.	Haemodialysis	34	Intervention 1 = 3.81 ± 4.3 years. Control = 3.12 ± 3.9 years.	Intradialytic resistance training programme and exercise counselling.	3 × week. Upper and lower body resistance training. Duration not reported.	Not reported.	Usual care	2 months	Hand Grip Strength	Not reported
Uchiyama et al., 2019 [40]	Japan	Parallel group RCT	Intervention = 64.9 ± 9.2 years; 79% male. Control = 63.2 ± 9.5 years; 70% male.	Peritoneal dialysis	47	Intervention 1 = 3.6 ± 2.7 years. Control = 4.0 ± 2.8 years.	Home-based exercise programme.	3 × week for 30 min at an exercise intensity 11–13 RPE. Upper and lower body resistance exercises.	52 ± 40% for aerobic exercise; 76 ± 37% for resistance exercise.	Usual care.	3 months	Hand Grip Strength	Yes
Umami et al., 2019 [41]	Indonesia	Parallel group RCT	Intervention 1 = 49.78 ± 11.65 years; 66.7% male. Intervention 2 = 46.38 ± 14.19 years; 53.8% male. Control = 50.54 ± 10.83 years; 46.2% male.	Haemodialysis	120	Intervention 1 = 48 (4, 192) months. Intervention 2 = 48 (6, 204) months. Control = 60 (5, 240) months. Data presented as median (IQR).	Intradialytic aerobic exercise programme and intradialytic aerobic and resistance exercise programme.	Intervention 1 = 2 × week for 30 min. Ergometer cycling at an intensity increasing to 60% to 80% HRmax. Intervention 2 = Lower body resistance training exercises. 3 × 10 repetitions.	Not reported	Usual care.	3 months	Gait Speed, Hand Grip Strength	Yes
Yeh et al., 2020 [42]	Taiwan	Parallel group RCT	Intervention = 57.87 ± 13.21 years; 63% male. Control = 53.91 ± 12.60 years; 47% male.	Haemodialysis	76	Intervention 1 = 63.47 ± 71.98 months. Control = 78.28 ± 63.95 months.	Intradialytic aerobic exercise programme	3 × week, for 30 min. Ergometer cycling at an intensity of RPE 12–14.	Not reported	Usual care.	3 months	STS60	Yes

Fat-free mass (FFM), lean body mass (LBM), mid-arm muscle area (MAMA), mid-arm muscle circumference (MAMC), mid-thigh muscle cross-sectional area (MT-CSA), randomised controlled trial (RCT), rating of perceived exertion (RPE), repetition max (RM), short physical performance battery (SPPB), sit-to-stand (STS), skeletal muscle mass (SMM), timed-up-and-go (TUG). Data are presented as mean ± SD unless otherwise stated.

**Table 2 nutrients-14-01817-t002:** Characteristics of trials containing either a nutritional or pharmacological intervention in the peritoneal dialysis and haemodialysis population.

Trial	Country	Trial Design	Participants	Haemodialysis or Peritoneal Dialysis	Sample Size (*n* = Randomised)	Dialysis Vintage	Type of Intervention	Intervention Description (Method of Delivery, Dose, Frequency, Duration)	Intervention Compliance	Type of Comparison	Length of Follow- Up	Sarcopenia- Related Outcomes	Prospective Power Calculation Reported
Ahmad et al., 1990 [43]	USA	Parallel group RCT	Intervention = 47.5 ± 2.5 years; 63% male. Control = 48 ± 2.4 years; 61% male. Data presented as mean ± SEM.	Haemodialysis	97	Intervention = 56.2 ± 6.6 months. Control = 60.7 ± 7.9 months. Data presented as mean ± SEM.	L-carnitine.	20 mg/kg of L-carnitine injected into the venous port of the blood circuit at the end of each dialysis session.	Not reported.	0.9% saline solution (placebo).	6 months	MAMA, MAMC	Not reported
Allman et al., 1990 [44]	Australia	Parallel group RCT	Intervention = 50 ± 11 years; 77.8% male. Control = 41 ± 18 years; 75% male.	Haemodialysis	32	Intervention = 40 ± 23 months. Control = 41 ± 28 months.	Water-soluble vitamin supplement.	A water-soluble vitamin supplement taken after each haemodialysis treatment.	Not reported.	Usual care (no placebo).	6 months	LBM	Not reported
Argani et al., 2014 [45]	Iran	Parallel group RCT	Intervention = 55.6 ± 4 years; 63% male. Control = 55.6 ± 8 years; 56% male.	Haemodialysis	66	Not reported.	Zinc sulphate.	A daily supplement of 440 mg of zinc sulphate in two divided doses for 60 days.	Not reported.	Placebo (corn starch) capsules.	60 days	FFM	Not reported
Brockenbrough et al., 2006 [46]	USA	Parallel group RCT	Intervention = 58.9 ± 14.9 years; 100% male. Control = 53.0 ± 17.2 years; 46% male.	Haemodialysis	40	Intervention = 43.6 ± 53.3 months. Control = 32.4 ± 47.2 months.	1% testosterone gel.	100 mg of topical 1% testosterone gel applied to the skin of the upper extremities or placebo for 6 months.	76% to 94% for the intervention and 61% to 84% for the placebo group.	Same as intervention but no active ingredient.	6 months	LBM	Yes
Calegari et al., 2011 [47]	Brazil	Parallel group RCT	Reported as total cohort = 56.4 ± 15.58; 83.3% male.	Haemodialysis	18	Reported as total cohort = 81.6 ± 36.76 years.	Oral nutritional supplement during each haemodialysis session.	3 × week. Oral nutritional supplement.	Not reported.	Not reported.	3 months	LBM	Not reported
Feldt-Rasmussen et al., 2007 [48]	Czech Republic, Denmark, France, Hong Kong, Israel, Poland, Singapore, Sweden & UK	Parallel group RCT	Intervention 1 = 58 ± 14 years; 62% male. Intervention 2 = 60 ± 15; 47% male. Intervention 3 = 61 ± 12; 62% male. Control = 59 ± 14 years; 68% male.	Haemodialysis	68	Intervention 1 = 48 ± 55 months. Intervention 2 = 42 ± 32 months. Intervention 3 = 26 ± 25 months. Control = 45 ± 62 months.	Daily subcutaneous injections of growth hormone.	Intervention 1 = 20 µg/kg per day. Intervention 2 = 35 µg/kg per day. Intervention 3 = 50 µg/kg per day.	Not reported.	Placebo injections.	6 months	Gait Speed, Hand Grip Strength, LBM	Yes
Fitschen et al., 2017 [49]	USA	Parallel group RCT	Intervention = 57 ± 8 years; 69% male. Control = 53 ± 13 years; 47% male.	Haemodialysis	41	Intervention = 43 ± 44 months. Control = 58 ± 35 months.	Beta-hydroxy-beta-methylbutyrate supplementation.	3 × a day; 1000 mg capsules of calcium beta-hydroxy-beta methylbutyrate.	5 participants in the intervention group were deemed noncompliant	Non-nutritive placebo capsule.	6 months	ALM, Gait Speed, LBM, STS30	Not reported
González-Espinoza et al., 2005 [50]	Mexico	Parallel group RCT	Intervention = 45.7 ± 14.4 years; 62% male. Control = 47.6 ± 17.4 years; 73% male	Peritoneal dialysis	30	Intervention = 20 (8, 35) months. Control = 15 (7.5, 24) months. Date presented as median (IQR).	Dried egg albumin-based supplement.	2 × day of 15 g of egg-based albumin supplement (equivalent of 11 g of high biological value protein).	90%	Usual care.	6 months	MAMA, MAMC	Not reported
Guida et al., 2019 [51]	Italy	Parallel group RCT	Intervention = 50.5 ± 11.5 years; 62% male. Control = 53.7 ± 10.6 years; 70% male.	Haemodialysis	23	Not reported.	Egg white dietary intervention.	3 × week; participants were instructed to replace one meal of the day with egg white.	Not reported.	Usual care	3 months	FFM	Yes
Hansen et al., 2000 [52]	Denmark	Parallel group RCT	Intervention = 44.4 ± 13 years; 55% male. Control = 48.3 ± 15 years; 64% male.	Haemodialysis	31	Intervention = 50 ± 43 months. Control = 71 ± 90 months.	Daily injection of growth hormone.	1 × day; administered by the participant at bedtime. At a dosage of 4 IU/mL.	Compliance was high, as only 1.7% of the total injections were missed.	Placebo consisted of freeze-dried glycine, mannitol, and sodium bicarbonate.	6 months	LBM	Not reported
Hewitt et al., 2013 [53]	Australia	Parallel group RCT	Intervention = 60 (53,71) months; 53% male. Control = 67 (54, 72) months; 43% male. Date presented as median (IQR).	Haemodialysis	60	Intervention = 38 (25, 66) months. Control = 42 (18, 89) months. Date presented as median [IQR].	Oral cholecalciferol.	1 × week, then 1 × month; 10 mL of an oral solution of medium-chain tri-glyceride containing 50,000 IU of cholecalciferol.	Not reported	Indistinguishable medium- chain triglyceride oral solution placebo.	6 months	Hand Grip Strength, STS5	Not reported
Hiroshige et al., 2001 [54]	Japan	Crossover RCT	Intervention = 75 ± 7 years; 43% male. Control = 74 ± 8 years; 50% male.	Haemodialysis	28	Intervention = 6.9 ± 3.1 years. Control = 6.8 ± 3.4 years.	Oral branch chained amino acid supplementation.	3 × day participant received oral branch chained amino acids at a total dose of 12 g per day.	100%	× 3 times a day. The placebo containing 6 g dextrose was identical in appearance and taste.	6 months	LBM	Not reported
Jeong et al., 2019 [22]	USA	Parallel group RCT	Intervention 1 = 56.6 ± 13 years; 51% male. Intervention 2 = 53.7 ± 11.4 years; 59% male. Control = 54.4 ± 12.3 years; 64% male.	Haemodialysis	138	Intervention 1 = 45.6 ± 38.7 months. Intervention 2 = 34.3 ± 34.8 months. Control = 47.9 ± 37.5 months.	Oral protein supplementation and intradialytic aerobic exercise programme.	Intervention 1 = 3 × week of 30 g of whey protein supplement. Intervention 2 = 3 × week of 30 g of whey protein and 45 min of ergometer cycling at RPE of 12–14.	>90% for study beverage and 80% exercise sessions.	Participants received 150 g of a non-nutritive beverage.	12 months	Gait Speed, LBM, STS30, TUG	Yes
Johannsson et al., 1999 [55]	Sweden	Parallel group RCT	Intervention = 73.5 ± 9 years; 70% male. Control = 72.7 ± 9 years; 70% male.	Haemodialysis	20	Not reported.	Post-dialysis subcutaneous injections of growth hormone.	3 × week. At a dose of 66.7 µg/kg (0.2 IU/kg of body weight).	Not reported	Indistinguishable placebo injections.	6 months	FFM, Gait Speed, Hand Grip Strength	Not reported
Johansen et al., 1999 [20]	USA	Parallel group RCT	Intervention = 44 ± 15 years; 79% male. Control = 50 ± 10 years; 80% male.	Haemodialysis & Peritoneal dialysis	29	Intervention = 2.9 ± 2.7 years. Control = 2.3 ± 2.0 years.	Intramuscular injection of nandrolone decanoate.	1 × week. At a dose of 100 mg/week.	Not reported.	Placebo injection of saline solution.	6 months	Hand Grip Strength, LBM	Not reported
Kopple et al., 2011 [56]	USA	Parallel group RCT	Intervention = 62 (26–96) years; 49% male. Control = 61 (19–95) years; 60% male. Data reported as mean (range).	Haemodialysis	712	Intervention = 4.2 (0.2–27.1) years. Control = 4.9 (0.3–34.6) years. Date presented as mean (range).	Injections of growth hormone.	Subcutaneous injections of growth hormone at a dose of 20 µg/kg/day.	Not reported.	Placebo injections.	104 weeks (terminated early, mean duration treatment = 20 weeks).	Hand Grip Strength, LBM	Yes
Kotzmann et al., 2001 [57]	Austria	Parallel group RCT	Intervention = 54.2 ± 14.3 years; 50% male. Control = 65.1 ± 11.4 years; 60% male.	Haemodialysis	19	Not reported.	Injections of growth hormone.	3 × week of 0.125 IU/kg (40.5 µg/kg) for the first four weeks and 0.25 IU/kg (81 µg/kg) thereafter.	Not reported.	Placebo injections.	3 months	LBM	Not reported
Li et al., 2020 [58]	China	Parallel group RCT	Intervention = 55.33 ± 10.11 years; 46% male. Control = 52 ± 12.3 years; 57% male.	Haemodialysis	29	Intervention = 6 (3, 9) years. Control = 3.5 (2, 6) years. Date presented as median (IQR).	Keto acid supplementation.	The intervention group supplemented with 0.1 g/kg/day of keto acid.	Not reported.	Usual care.	6 months	Gait Speed, Hand Grip Strength, LBM	Not reported
Luo et al., 2020 [59]	China	Parallel group RCT	Intervention = 55.8 ± 13.4 years; 52.9% male. Control = 55.3 ± 13.2 years; 56.7% male.	Peritoneal dialysis	142	Intervention = 3–12 months (*n* = 15), 12–26 months (*n* = 37), >36 months (*n* = 16). Control = 3–12 months (*n* = 18), 12–26 months (*n* = 34), >36 months (*n* = 15).	Nurse led personalised dietary plans.	Personal dietary plans based on the food exchange models.	Not reported.	Usual care.	12 months	MAMC	Not reported
Marini et al., 2020 [60]	Brazil	Parallel group RCT	Intervention 1 = 41.86 ± 3.32 years; 71% male. Control = 41.79 ± 2.72 years; 64% male. Data presented as mean ± SEM.	Haemodialysis	30	Not reported.	Creatine supplementation.	(5 g) 4 × day for week 1 (loading period) and then 1 × day for 2–4 weeks.	Not reported.	10 g of maltodextrin (placebo).	1 month	Gait Speed, LBM	Yes
Maruyama et al., 2019 [18]	Japan	Parallel group RCT	Intervention = 72 ± 9 years; 42% male. Control = 72 ± 10 years; 42% male.	Haemodialysis	91	Intervention = 79 ± 47 months. Control = 74 ± 47 months.	L-carnitine supplementation.	3 × week; injections of L-carnitine (1000 mg) after each dialysis session.	Not reported.	Usual care.	12 months	Hand Grip Strength, LBM, MAMA, SMM	Yes
Sahathevan et al., 2018 [61]	Malaysia	Parallel group RCT	Intervention = 50.84 ± 15.20 years; 45.9% male. Control = 42.14 ± 14.57 years; 40.5% male.	Peritoneal dialysis	126	Intervention = 3.27 ± 3.03 years. Control = 3.19 ± 2.59 years.	Whey protein supplementation.	2 × day of 15 g whey protein sachets.	Compliance for the intervention was 75 ± 18%.	Usual care.	6 months	Hand Grip Strength, LBM, MAMA, MAMC	Yes
Schincaglia et al., 2020 [62]	Brazil	Parallel group RCT	Intervention = 49.3 ± 3.4 years; 66.6% male. Control = 51.3 ± 3 years; 64.7%.	Haemodialysis	43	Not reported	Baru almond oil supplementation.	10 × day; 500 mg capsules of Baru oil each day.	Not reported.	Capsules of mineral oil placebo.	3 months	FFM	Yes
Supasyndh et al., 2013 [63]	Thailand	Parallel group RCT	Intervention = 41.0 ± 10.5 years; 52.6% male. Control = 45.1 ± 8.5 years; 68.2% male.	Haemodialysis	43	Intervention = 98 (61, 110) months. Control = 96 (59, 115.7) months. Date presented as median (IQR).	Oxymetholone	2 × day of oxymetholone 50 mg orally.	Not reported.	Received a placebo that was identical in appearance to the active drug.	6 months	FFM, Hand Grip Strength	Not reported
Singer et al., 2019 [64]	Australia	Parallel group RCT	Intervention = 59.5 ± 15.6 years; 64% male. Control = 63.8 ± 14.2 years; 72% male	Haemodialysis & Peritoneal dialysis	68	Intervention = 21.7 (5.3, 54.9] months. Control = 7.6 [3.7, 43.1]. Date presented as median [IQR].	Cholecalciferol supplementation	1 × week of a capsule containing 50,000 U of cholecalciferol. Study dose was adjusted at 3 and 6 months.	Adherence reported as excellent.	Placebo capsules.	12 months	Hand Grip Strength	Yes
Teixido-Planas et al., 2005 [65]	Spain	Parallel group RCT	Intervention = 56.57 (13, 22) years; 57% male. Control = 58.43 (14, 63) years; 56% male. Data reported as mean (range).	Peritoneal dialysis	65	Not reported.	Oral protein supplement.	× 1 a day of a 200 mL oral protein drink.	Not reported.	Usual care.	12 months	LBM, MAMC	Yes
Tomayko et al., 2015 [66]	USA	Parallel group RCT	Intervention 1 = 57 ± 4.8 years; 63.6% male. Intervention 2 = 52.5 ± 4.3 years; 58.3% male. Control = 53.3 ± 2.4; 66.7% male.	Haemodialysis	46	Not reported.	Oral protein supplement.	Intervention 1 = × 3 a week. 27 g of whey protein provided during dialysis. Intervention 2 = × 3 a week. 27 g of soy protein during dialysis.	A level of 75% compliance was established for the study.	A non-caloric placebo powder during dialysis.	6 months	Gait Speed, LBM, STS30, TUG	Not reported
Wu et al., 2011 [67]	Taiwan	Parallel group RCT	Intervention = 45.2 ± 12.9 years; 37% male. Control = 40.5 ± 12.9 years; 36% male	Peritoneal dialysis	44	Intervention = 4 ± 2.2 years. Control = 3.1 ± 2.7 years.	L-carnitine supplementation	1 × day of a 600 mg oral L-carnitine tablet.	Not reported.	Usual care (no placebo).	6 months	Hand Grip Strength, MAMA, MAMC	Not reported
Wu et al., 2015 [68]	USA	Parallel group RCT	Intervention = 52.6 ± 3.3 years; 61.5% male. Control = 55.9 ± 2.6 years; 64.3% male.	Haemodialysis	33	Intervention = 75.5 ± 14.1 months. Control = 59.8 ± 10.6 months.	Pomegranate extract supplementation	1 × day of 1000 mg oral capsule containing purified pomegranate polyphenol extract.	95.9% and 98.2% for the intervention and placebo groups respectively.	A non-caloric placebo capsule.	6 months	STS30, TUG, 1 RM	Not reported

Fat-free mass (FFM), lean body mass (LBM), mid-arm muscle area (MAMA), mid-arm muscle circumference (MAMC), randomised controlled trial (RCT), repetition max (RM), sit-to-stand (STS), skeletal muscle mass (SMM), timed-up-and-go (TUG). Data are presented as mean ± SD unless otherwise stated.

**Table 3 nutrients-14-01817-t003:** Characteristics of trials in the transplant recipient population reporting an outcome associated with sarcopenia.

Trial	Country	Trial Design	Participants	Sample Size (*n* = Randomised)	Dialysis Vintage	Type of Intervention	Intervention Description (Method of Delivery, Dose, Frequency, Duration)	Intervention Compliance	Type of Comparison	Length of Follow- Up	Sarcopenia- Related Outcomes	Prospective Power Calculation Reported
Greenwood et al., 2015 [69]	UK	Parallel group RCT	Intervention 1 = 53.9 ± 10.7 years; 77% male. Intervention 2 = 54.6 ± 10.6 years; 54% male. Control = 49.5 ± 10.6 years; 50% male.	60	Not reported.	Aerobic and resistance exercise programme.	Intervention 1 = 3 × week aerobic training. Treadmill running, cycling, and elliptical training at an RPE 13–15 for 60 min. Intervention 2 = × 3 upper and lower body resistance exercises for 60 min.	87.4 ± 5.2%.	Usual care.	3 months	STS60	Not reported
Henggeler et al., 2018 [70]	New Zealand	Parallel group RCT	Intervention = 49.2 ± 14.6 years; 66% male. Control = 48.3 ± 13.9 years; 72% male.	37	Not reported.	Lifestyle intervention (physical activity and nutritional counselling).	× 8 additional consultations with a dietitian, physical activity and exercise advice at 2, 3, and 6 months post-transplant.	93% for intervention; 97% for control.	Usual care.	12 months	FFM, Gait Speed, Hand Grip Strength, LBM, MAMA	Yes
Hernández Sánchez et al., 2021 [71]	Spain	Parallel group RCT	Intervention = 49.7 ± 9.6 years; 37.5% male. Control = 48.6 ± 10.6 years; 75% male.	16	Intervention = 115 ± 54 months. Control = 88 ± 53 months.	Resistance exercise programme.	2 × week. For 60 min. Walking plus upper and lower body resistance training.	100%.	Usual care.	2.5 months	Hand Grip Strength, STS60, TUG	Not reported
Karelis et al., 2016 [72]	Canada	Parallel group RCT	Intervention = 45.3 ± 14 years; 60% male. Control = 39.4 ± 8 years; 60% male.	24	Not reported.	Resistance exercise programme.	3 × week. For 45–60 min. Upper and lower body resistance exercises.	80%.	Usual care.	4 months	LBM	Not reported
Lima et al., 2021 [73]	Brazil	Parallel group RCT	Intervention = 54 ± 3 years; 43% male. Control = 43 ± 18 years; 0% male.	41	Intervention = 4 ± 1 years. Control = 4 ± 2 years.	Aerobic and resistance exercise programme	3 × week. 30 min of aerobic cycling and upper and lower body resistance exercises.	Not reported.	Usual care.	4 months	Hand Grip Strength, LBM	Yes
Painter et al., 2002 [74]	USA	Parallel group RCT	Intervention = 39.7 ± 12.6 years; 55.5% male. Control = 43.7 ± 10.7 years; 69.1% male.	167	Not reported.	Aerobic exercise programme.	4 × week. Primarily walking or cycling exercise. 30 m mins duration.	Not reported.	Usual care.	12 months	LBM	Not reported
Painter et al., 2003 [75]	USA	Parallel group RCT	Intervention = 48.3 ± 12.7 years; 66% male. Control = 46.8 ± 14.4 years; 78% male.	36	Not reported.	Early steroid withdrawal.	Participants randomised into rapid elimination of steroids were decreased to 30 mg at day 4 and were withdrawn at day 5.	Not reported.	Usual care.	12 months	LBM	Not reported
Riess et al., 2014 [76]	Canada	Parallel group RCT	Intervention = 56.9 ± 12.2 years; 50% male. Control = 52.4 ± 14.3 years; 40% male.	31	Intervention = 6.4 ± 4.1 years. Control = 9.1 ± 8.8 years.	Aerobic and resistance exercise programme.	2 × week. Cycling and treadmill training for 30–60 min at 60–80 VO^2^ peak. Lower body resistance training.	81%	Usual care.	3 months	LBM	Not reported
Tzvetanov et al., 2014 [77]	USA	Parallel group RCT	Intervention = 46.9 ± 6.9 years; 50% male. Control = 45 ± 19 years; 37.5% male.	17	Intervention = 8.6 ± 6.2 months. Control = 10.9 ± 7.6 years.	Lifestyle intervention (resistance training and nutritional support).	2 × week. Resistance exercise sessions. Duration not reported. Cognitive behavioural therapy and nutritional support.	100% adherence in the intervention group.	Usual care.	12 months	LBM	Not reported.
van den Ham et al., 2003 [78]	Netherlands	Parallel group RCT	Intervention 1 = 56.3 ± 17.2 years; 70% male. Control = 52.4 ± 13.6 years; 82% male.	27	Not reported.	Early steroid withdrawal.	Participants in the intervention group were withdrawnfrom steroids within 2 weeks.	Not reported.	Usual care.	6 months	LBM	Not reported.

Fat-free mass (FFM), lean body mass (LBM), mid-arm muscle area (MAMA), randomised controlled trial (RCT), sit-to-stand (STS), timed-up-and-go (TUG). Data are presented as mean ± SD unless otherwise stated.

**Table 4 nutrients-14-01817-t004:** The effect of exercise programmes outside of haemodialysis treatment on grip strength.

	Intervention		Control	
Trial	Baseline	Follow-up	Baseline	Follow-up
Frih et al., 2017 [26]	29.8 ± 6 N (*n* = 21)	37.4 ± 4.8 N (*n* = 21)	29.3 ± 5.6 N (*n* = 20)	30 ± 5.2 N (*n* = 20)
Koh et al., 2010 [29]	36 ± 15 kg (*n* = 14) (home-based)	37 ± 14 kg (*n* = 14) (home-based)	28 ± 13 kg (*n* = 14)	31 ± 12 kg (*n* = 14)
Rosa et al., 2021 [35]	23 ± 6 kg (*n* = 55) (Dynamic training group); 25 ± 5 (*n* = 51) (isometric training group)	35 ± 4 kg (*n* = 55) (Dynamic training group); 38 ± 7 (*n* = 51) (isometric training group)	24 ± 8 kg (*n* = 52)	26 ± 5 kg (*n* = 52)
Song et al., 2012 [37]	26.3 ± 8.5 kg (*n* = 20)	28.7 ± 9 kg (*n* = 20)	26.2 ± 10.2 kg (*n* = 20)	27.8 ± 11.8 kg (*n* = 20)

Data are reported as mean ± SD.

## Data Availability

Data will be made available upon a request to the corresponding author.

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
