# Peer review of "The Effect of Non-Pharmacological and Pharmacological Interventions on Measures Associated with Sarcopenia in End-Stage Kidney Disease: A Systematic Review and Meta-Analysis"

_nutrients, 2022, doi:10.3390/nu14091817_

Round 1

Reviewer 1 Report

The authors performed a comprehensive systematic review and meta-analysis of currently available studies investigating the effects of pharmacological, nutritional or exercise interventions on sarcopenia-related measures in patients with ESKD (including transplant recipients).

The paper is interesting and well written. Both search strategy and data analysis are sound. The results are clearly presented and the conclusions perfectly summarize the main findings of the study. Possible future research goals have been properly highlighted. 

In my opinion the manuscript is suitable for publication after minimal editing.

Author Response

Response to Reviewer Comments 1

We thank Reviewer 1 for reviewing our manuscript and for their encouraging comments.

Reviewer 2 Report

This is a well performed and written systematic review and meta-analysis. Sarcopenia is an important problem in CKD, and this paper nicely presents the best available evidence for interventions against sarcopenia at the moment. I have few suggestions to improve the paper.

The exercise interventions were separated into intradialytic and outside dialysis, and between HD, PD and TX. Would it be inappropriate to pool some of these data, and if so please express why.

Page 2: 2.4 Comparison: The second sentence is difficult to understand. Could you please rephrase

The date when the literature search was performed should be provided. All search strings should be provided in the appendix, not only an example.

Were any attempts performed to reach out for results from unpublished material/grey literature, such contact to authors? 

Was snowballing performed?

It would be a help to the reader if all the forest plots had the control/experimental on the same side of the 0-line

How many of the studies included patients without sarcopenia? Could this be the reason why they did not find any effect of the intervention?

Was there any publications bias, especially in the exercise groups on the primary outcomes? Could Funnel plots be provided?

I suggest that possible side effects to the interventions/treatments also are mentioned in the discussion and should be explored in future trials. 

Author Response

Response to Reviewer Comments 2

We would like to thank Reviewer 2 for their careful evaluation of our initial submission and providing us with specific comments and suggestions which we feel have significantly improved the overall quality of our manuscript. Please find our responses to all comments below. All reviewer comments are in bold with our responses below. We have highlighted all changes based on the reviewer comments within the manuscript in red.

The exercise interventions were separated into intradialytic and outside dialysis, and between HD, PD and TX. Would it be inappropriate to pool some of these data, and if so please express why.

Thank you for this suggestion. Whilst we do partly agree that pooling some of the data for the specific end-stage kidney disease (ESKD) subpopulations may be attractive, we do not feel that in this case it is appropriate. In the present review the pooling of data would be only possible for the exercise/physical activity interventions. However, the majority of the exercise and/or physical activity interventions for the haemodialysis population take place during dialysis (intradialytic) we do not feel that it is appropriate to pool these with other heterogeneous interventions such as (for example) programmes of walking taking place outside of dialysis. This will likely introduce heterogeneity into the meta-analyses, an example of this is for the four exercise trials measuring the effect of (Frih et al., 2017, Koh et al., 2010, Rosa et al., 2021, Song et al., 2012) programmes of exercise taking place outside of dialysis on hand grip strength, when this data is pooled the I2 value is 89%, making it unsuitable for meta-analysis. There are also subtle differences within these subpopulations that could influence the effectiveness of exercise interventions This is highlighted in a paper by Wilkinson et al. (https://pubmed.ncbi.nlm.nih.gov/31725147/) which showed that transplant recipients are significantly more active compared to individuals receiving haemodialysis and peritoneal dialysis. Lastly were we to pool some of these data it may be misleading, as it will give the impression that there is good evidence for exercise and or physical activity interventions in individuals receiving peritoneal dialysis or transplant recipients, when in fact there is a lack of randomised controlled trial data for these populations.

Page 2: 2.4 Comparison: The second sentence is difficult to understand. Could you please rephrase

Thank you for pointing this out. We have now rephrased this sentence (please see lines 86-88). We now hope that it is to the Reviewers satisfaction.

The date when the literature search was performed should be provided. All search strings should be provided in the appendix, not only an example.

Thank you for this comment. We have now included the exact date when the searches were performed (see line 116). We have provided the search strings for MEDLINE, EMBASE and CINAHL. The other databases were search using different combinations of these search terms. We have clarified this in lines 118-120.

Were any attempts performed to reach out for results from unpublished material/grey literature, such contact to authors? 

We specifically searched the databases Open Grey and Conference Proceedings Citation Index (Web of Science) in attempt to identify Grey Literature. We contacted members of the Physical Activity and Wellbeing Kidney Research UK Clinical Study Group in the United Kingdom in an attempt to identify eligible trials. We have added this to lines 117-120.

Was snowballing performed?

Snowballing was not performed. We are happy to clarify this in section 2.7 Search Strategy should the Editor deem this appropriate.

It would be a help to the reader if all the forest plots had the control/experimental on the same side of the 0-line

We though carefully about the most appropriate way in which to present the data in the forest plots. Whilst we agree that is would help the reader to have “favours (control)” and “favours (experimental)” on the same side of the line for all the forest plots we do not believe that this is appropriate. The reason for this is due to the values for the outcome measures, for example for the sit-to-stand 5, a negative score favours the experimental (as a quicker score in seconds is “better” for this outcome measure). While for the sit-to-stand 60 a positive score (or a higher number of repetitions) favours the experimental as this indicates a “better” score for this outcome measure. To summarise which side of the 0 line the “favours (control)” and “favours (experimental)” is entirely dependent on how the outcome measure is assessed, and therefore we do not deem it appropriate to change this. For nearly all of the outcome measures (with the exception of the sit-to-stand 5 and the timed-up-and go) a positive score favours the experimental.

How many of the studies included patients without sarcopenia? Could this be the reason why they did not find any effect of the intervention?

For the included studies there were no specific inclusion/exclusion criteria for sarcopenia. However, in the end-stage kidney disease (ESKD) population it has been estimated that sarcopenia may be present in up to 37% of the population (as outlined in the introduction). Given this number it is reasonable to assume that many of the included trials included individuals with sarcopenia. This review demonstrates that exercise is an effective intervention in modifying measures associated with sarcopenia in the ESKD population (this is in agreement with other reviews and with the recent consensus paper (https://pubmed.ncbi.nlm.nih.gov/30312372/) which highlights physical exercise as a key component of the treatment for sarcopenia). There was a lack of evidence for nutritional and pharmacological interventions in this population. However, it is unlikely that nutritional interventions alone (without the exercise stimulus) would improve measures associated with sarcopenia. For the pharmacological interventions there was only a small number of trials performed in this population. In summary we do not believe that the lack of effect of nutritional and pharmacological interventions is a result of the identified trials including individuals without sarcopenia. At present the strongest therapeutic intervention for sarcopenia is exercise (which this review supports).

Was there any publications bias, especially in the exercise groups on the primary outcomes? Could Funnel plots be provided?

Thank you for this suggestion. We have now included funnel plots (as Appendix B) for all the meta-analyses performed for the primary outcome (muscle strength). Please see lines 139-141, 201 and Appendix B. Formal testing was not performed to test for plot asymmetry as none of the analyses contained more than ten trials (as recommended by the Cochrane Collaboration Handbook). After visual inspection of the funnel plots there was no evidence of publication bias.

I suggest that possible side effects to the interventions/treatments also are mentioned in the discussion and should be explored in future trials. 

Thank you for this comment. We have added that the available evidence in the literature suggests that intradialytic exercise is safe (please see line 416). We have also highlighted the need for future trials to assess the safety of pharmacological and nutritional interventions (please see lines 440-442).
